# Improving public understanding of 'overdiagnosis' in England: a population survey assessing familiarity with possible terms for labelling the concept and perceptions of appropriate terminology

Alex Ghanouni, Cristina Renzi, Jo Waller

Research Department of Behavioural Science and Health, University College London, London, UK

**Correspondence to**
Dr Jo Waller; j.waller@ucl.ac.uk

## ABSTRACT

**Objectives** Communicating the concept of 'overdiagnosis' to lay individuals is challenging, partly because the term itself is confusing. This study tested whether alternative descriptive labels may be more appropriate.

**Design** Questionnaire preceded by a description of overdiagnosis.

**Setting** Home-based, computer-assisted face-to-face survey.

**Participants** 2111 adults aged 18–70 years in England recruited using random location sampling by a survey company. Data from 1888 participants were analysed after exclusions due to missing data.

**Interventions** Participants were given one of two pieces of text describing overdiagnosis, allocated at random, adapted from National Health Service breast and prostate cancer screening leaflets.

**Primary and secondary outcome measures** Main outcomes were which of several available terms (eg, 'overdetection') participants had previously encountered and which they endorsed as applicable labels for the concept described. Demographics and previous exposure to screening information were also measured. Main outcomes were summarised with descriptive statistics. Predictors of previously encountering at least one term, or endorsing at least one as making sense, were assessed using binary logistic regression.

**Results** 58.0% of participants had not encountered any suggested term; 44.0% did not endorse any as applicable labels. No term was notably familiar; the proportion of participants who had previously encountered each term ranged from 15.9% to 28.3%. Each term was only endorsed as applicable by a minority (range: 27.6% to 40.4%). Notable predictors of familiarity included education, age and ethnicity; participants were less likely to have encountered terms if they were older, not white British or had less education. Findings were similar for both pieces of information.

**Conclusions** Familiarity with suggested terms for overdiagnosis and levels of endorsement were low, and no clear alternative labels for the concept were identified, suggesting that changing terminology alone would do little to improve understanding, particularly for some population groups. Explicit descriptions may be more effective.

## Strengths and limitations of this study

► This study assessed (predictors of) familiarity and perceived appropriateness of a broad range of possible terms for the concept of 'overdiagnosis' among a large, representative sample of the general public in England.

► The concept was described based on information that was widely used by the National Health Service in England, maintaining generalisability.

► The demographic criteria used to achieve population representativeness were not comprehensive and response rates were not available.

► The list of possible terms was also not exhaustive; more suitable alternative labels may have been omitted.

► Results may be time dependent; ongoing communication initiatives may change public perceptions of appropriate terms for the concept of 'overdiagnosis'.

## INTRODUCTION

There is growing concern among healthcare providers and policymakers about the potential for medical tests to detect asymptomatic disease that would never have become clinically apparent or resulted in death.[1] 'Overdiagnosis' can harm, primarily via the subsequent risks and costs associated with unnecessary treatment and also via opportunity costs to a healthcare service due to overuse of scarce resources.[2] Various initiatives have aimed to propagate understanding of the concept among patients and the public so that they can make better informed decisions about their health (eg, refs [3–5]). However, previous studies have found that the concept is challenging to communicate to lay people (eg, in the context of breast cancer screening).[6 7]

Little is known about how to improve communication of overdiagnosis.[8] One barrier is that the term 'overdiagnosis' itself

BMJ

may be confusing and counterintuitive. For example, a previous study in the UK asked members of the public to give their interpretation of what the word meant; a large proportion of participants gave definitions that did not match the concept but resembled other similar words (eg, 'providing an incorrect diagnosis' since 'overdiagnosis' resembles 'misdiagnosis').[9] Equivalent findings have been reported by a previous survey in Australia.[10] In addition, a focus group study undertaken as part of the development of a decision aid for breast screening found that 'overdiagnosis' was not understood intuitively and that 'overdetection' may be clearer.[11]

These findings suggest that the term 'overdiagnosis' does not clearly reflect its intended meaning for the general public and that other terms (such as 'overdetection') may be more appropriate. Hence, this study asked a large sample of the public in England to appraise descriptions of the concept and tested which of several possible terms people thought made sense as labels (ie, endorsement). Alternatives included 'overdetection' or 'unnecessary diagnosis'[11] as well as terms related to unnecessary therapy (eg, 'overtreatment') and terms that were less technically accurate but still had the potential to be seen as applicable (eg, 'false positives'[9 10]). A limitation to the utility of 'overdiagnosis' as a label may be the currently low level of public awareness of the term.[9 12] Consequently, this study also measured which terms people had encountered before (ie, familiarity) to test whether any alternatives were notably more familiar. We also explored whether participant characteristics were associated with having previously encountered these terms, or with endorsing them as making sense, as this may aid targeted communication efforts.

Information was derived from materials that have been widely used by the National Health Service (NHS) Breast Screening Programme[13] and the Prostate Cancer Risk Management Programme.[14] The study was carried out as part of wave 3 of the Attitudes, Behaviour and Cancer UK Survey (ABACUS). Other studies arising from this survey have also been published (eg, ref [15]).

## METHODS
### Design
The focus of the present study was on appropriate terminology for widely used descriptions of overdiagnosis. However, the ABACUS survey was a broader piece of work that also compared the perceived clarity of different forms of overdiagnosis information. Hence, elements of the design and measures have also been reported elsewhere.[15] Face-to-face computer-assisted interviews were conducted by TNS (a market research company) as part of a weekly omnibus survey and took place in participants' homes between April and May 2016. Participants were informed:

> We would now like to ask you some questions about leaflets on health-related topics. The NHS offers people a variety of screening tests to check for illnesses before symptoms have appeared. People offered an

NHS screening test are often given a leaflet that explains the risks and benefits of having the test. The leaflet is either posted or given out by a doctor or nurse.

Participants then received information on overdiagnosis adapted from written material used by the NHS in England as part of either (1) the Breast Screening Programme or (2) the Prostate Cancer Risk Management Programme as they existed in February 2016 (neither of which used a specific label), allocated at random in a 1:1 ratio:

> The test can find an illness that would never have caused a person harm. Some people will be diagnosed and treated for an illness that would never otherwise have been found and would not have become life-threatening. (From the breast screening information leaflet)[13]

> The test may make you worry by finding an illness that may never cause any symptoms or shorten your life. (From the prostate screening information leaflet)[14]

The Breast Screening Programme in England offers triennial mammography to all women between the ages of 50 years and 70 years and registered with a general practitioner (GP). The Prostate Screening Risk Management Programme allows men aged 50 years or older to have a Prostate Specific Antigen test by requesting it from their GP after discussing the risks and benefits. References to cancer were removed so that findings could be generalised to beyond these two specific screening contexts.

### Participants
Participants were members of the public in England aged 18–70 years. National representativeness was sought via a two-stage process. In the first instance, random location sampling was carried out using the Postcode Address File and Census statistics. Within each selected location, participant quotas were set based on demographic characteristics (ie, age, gender, employment status and presence of children in the home).

### Measures
#### Demographics
Relevant measures consisted of age, gender, ethnicity, social grade,[16] marital status, highest level of education obtained, personal history of cancer ('*Have you ever been diagnosed with cancer?*': '*yes*' or '*no*'), and whether anyone they know had been diagnosed with cancer ('*Has anyone close to you ever been diagnosed with cancer?*': '*yes*', '*no*' or '*don't know*').

#### Familiarity with screening information
Participants were asked whether they had ever previously received information about screening (for cancer or other illnesses) via (1) reading a leaflet from the NHS, (2) reading an NHS website and (3) talking with a doctor or nurse (response options: '*yes*', '*no*' and '*not sure*').

If participants were eligible for cancer screening and had not been diagnosed with the target disease, they were asked about their previous participation (eg, only women aged

47–70 years without a breast cancer diagnosis were asked about breast screening participation) via items designed using the Precaution Adoption Process Model,[17] a stage theory of protective health behaviour. This measures whether participants have heard of a particular type of screening, whether they have heard of a type of screening but have never been invited, whether they have been invited but never participated, whether they have participated but not consistently and whether they have participated consistently. Eligibility criteria and exact question wording for assessing experience have been reported elsewhere.[15]

### Perceived clarity of information

After reading the information allocated at random, participants were asked, '*How clear do you find this description of a risk of the test?*' with available responses of '*extremely clear*', '*very clear*', '*moderately clear*', '*slightly clear*' and '*not at all clear*'. This was followed by a question on whether they had read or heard similar information about a screening test before (possible responses were: '*yes*', '*no*' and '*not sure*').

### Main outcomes

The first main outcome measure (endorsement) consisted of '*Does this term make sense to you as a way of describing this risk of a screening test?*', followed by seven terms: '*overdiagnosis*', '*overdetection*', '*unnecessary diagnosis*', '*overtreatment*', '*unnecessary treatment*', '*false positive diagnosis*' and '*false positive test results*'[i] (responses: '*yes*' or '*no*'). Suggested terms were based on previous literature (eg, ref [11]) and discussion among the research team.

The second outcome measure (familiarity) was: 'Have you ever seen or heard any of the previous seven terms before today?' and if participants responded '*yes*', they were asked which out of the previous terms (responses: '*yes*', '*no*' or '*not sure*').

### Ancillary measures

The measures used in this study were part of the broader ABACUS survey. Most of the items in this study were presented at the start of the survey and so the effects of priming from unrelated items were expected to be minimal. In order, the full survey included: questions on decision-making style and previous exposure to cancer screening information; information on overdiagnosis adapted from screening leaflets; questions on perceived clarity of this information, previous exposure to similar information, the main study outcomes and help-seeking behaviour in relation to cancer screening, followed by more questions on self-rated health; questions on cancer diagnoses, perceived cancer risk, screening behaviour in relation to each of the applicable programmes (cervical, breast, bowel and prostate) and educational level.

### Public involvement

Public involvement consisted of input into the design of the study and development of measures via pilot testing of the survey. This comprised two stages: first, a series of telephone cognitive interviews[18] was used to determine whether items could be understood by lay people (n=11) and that the survey was not overly burdensome. Participants were asked for feedback on any items they found difficult to understand or answer, and the survey was revised accordingly. Second, a web-based version of the survey was pilot-tested with 431 participants. There are currently no plans to disseminate the results to study participants.

### Analysis

Participants were excluded if they declined to answer any of the measures included in this study. Ethnicity and marital status were dichotomised into 'white British' and 'other ethnic groups', and 'single, widowed, divorced or separated' and 'married or living as a couple', respectively. Social class grade was categorised as 'grades A or B', 'grades C1 or C2' and 'grades D or E'. Education was coded based on levels 1–4 from the Office of National Statistics[19] or 'Other' for non-ordinal levels of education such as professional qualifications.

Sample characteristics were summarised with descriptive statistics. We report the percentages of participants (with binomial 95% CIs) who had (1) previously encountered and (2) endorsed each possible number of terms (from 0 to 7 in both cases), and the percentages of participants who had previously encountered and endorsed each specific term.

Responses to (1) and (2) were also recoded into 'previously seen or heard one or more terms' (familiarity) and 'endorsed one or more terms as making sense to them' (endorsement) for further evaluation, for example, 'was familiar with at least one term' versus 'was not familiar' (including responses of 'not sure'). Two exploratory logistic regression models tested the null hypothesis that predictor variables were unrelated to either of these outcomes. In both models, predictor variables consisted of demographic characteristics, familiarity with screening information, the overdiagnosis information condition (breast vs prostate) and perceived clarity. In addition, each model included the outcome from the other as a predictor (eg, the endorsement model included familiarity as a potential predictor variable). Variance inflation factors were small (all <2.969 and <2.972, respectively), indicating that (multi)collinearity was limited, and a Box-Tidwell procedure found little evidence to suggest that the age variable violated the assumption of linearity (p values: 0.110 and 0.508). Adjusted ORs, accompanying 95% CIs and p values for having previously seen or heard one or more terms before and endorsing one or more term as making sense are reported alongside descriptive statistics.

### RESULTS

### Participant characteristics

After excluding 223 participants with missing data, 1888 cases were analysed. Sample characteristics are presented in table 1. Mean age was 43.6 years (SD: 15.7).

---

[i] Participants were also asked, '*Can you think of any other terms to describe this risk that would make sense to you?*'. However, only approximately 60 responses suggesting potentially applicable terms were recorded across all participants (eg, '*oversensitivity*', '*unknown diagnosis*'), of which most were mentioned just once and so no further analysis was attempted.

**Table 1**    Characteristics of the sample

| Characteristic | n | (%) |
|---|---|---|
| **Overdiagnosis information** | | |
| Breast screening text | 987 | (52.3) |
| Prostate screening text | 901 | (47.7) |
| **Gender** | | |
| Male | 879 | (46.6) |
| Female | 1009 | (53.4) |
| **Ethnicity** | | |
| White British | 1432 | (75.8) |
| Other ethnic groups | 456 | (24.2) |
| **Marital status** | | |
| Married or living as a couple | 1133 | (60.0) |
| Single, widowed, divorced or separated | 755 | (40.0) |
| **Highest level of education\*** | | |
| No formal qualifications | 281 | (14.9) |
| Approximately level 1, 2 or 3 | 890 | (47.1) |
| Approximately level 4 | 507 | (26.9) |
| Other | 199 | (10.5) |
| Don't know/not sure | 11 | (0.6) |
| **Social class grade†** | | |
| Grade A or B | 386 | (20.4) |
| Grade C1 or C2 | 931 | (49.3) |
| Grade D or E | 571 | (30.2) |
| **Personal diagnosis of cancer** | | |
| Yes | 98 | (5.2) |
| No | 1790 | (94.8) |
| **Knows someone with cancer** | | |
| Yes | 1113 | (59.0) |
| No | 771 | (40.8) |
| Don't know/not sure | 4 | (0.2) |
| **Any previous cervical screening experience** | | |
| Yes | 652 | (34.5) |
| No | 226 | (12.0) |
| Not eligible | 1010 | (53.5) |
| **Any previous breast screening experience** | | |
| Yes | 317 | (16.8) |
| No | 83 | (4.4) |
| Not eligible | 1488 | (78.8) |
| **Any previous bowel screening experience** | | |
| Yes | 293 | (15.5) |
| No | 181 | (9.6) |
| Not eligible | 1414 | (74.9) |

Continued

**Table 1**    Continued

| Characteristic | n | (%) |
|---|---|---|
| **Any previous prostate screening experience** | | |
| Yes | 77 | (4.1) |
| No | 344 | (18.2) |
| Not eligible | 1467 | (77.7) |
| **Previously read a screening leaflet** | | |
| Yes | 1058 | (56.0) |
| No | 809 | (42.8) |
| Don't know/not sure | 21 | (1.1) |
| **Previously read an NHS screening website** | | |
| Yes | 324 | (17.2) |
| No | 1549 | (82.0) |
| Don't know/not sure | 15 | (0.8) |
| **Discussed screening with doctor/nurse** | | |
| Yes | 624 | (33.1) |
| No | 1251 | (66.3) |
| Don't know/not sure | 13 | (0.7) |
| **Previously read or heard similar information** | | |
| Yes | 662 | (35.1) |
| No | 1185 | (62.8) |
| Don't know/not sure | 41 | (2.2) |
| **Perceived clarity of information** | | |
| Extremely clear | 170 | (9.0) |
| Very clear | 636 | (33.7) |
| Moderately clear | 685 | (36.3) |
| Slightly clear | 206 | (10.9) |
| Not at all clear | 191 | (10.1) |

\*Level 1–3 qualifications include, for example, GCSEs and A Levels; level 4 qualifications include degrees and higher degrees.
†Social grades are based on occupation (eg, grade A includes managerial roles; grade E includes casual workers).
GCSEs, General Certificate of Secondary Education; NHS, National Health Service.

## Familiarity with terms

Table 2 illustrates that the majority of participants (58.0%) were not familiar with any of the listed terms. However, the second most common number of terms recognised as familiar was all seven (12.3%). The percentages of participants previously encountering between one and six terms ranged from 1.1% to 9.4%. Table 2 also shows that the percentages of participants who had encountered each specific term ranged from 15.9% ('*overdetection*') to 28.3% ('*false positive test results*').

Table 3 reports predictors of familiarity with at least one term. Among demographic characteristics, there was strong evidence against the null hypothesis for ethnicity,

**Table 2** Number of participants (and percentages and 95% CIs) previously encountering (1) each number of possible terms and (2) each specific term

| Screening information | Total | Number of terms previously encountered: n, %, 95 % CIs | | | | | | | |
|---|---|---|---|---|---|---|---|---|---|
| | | 0 | 1 | 2 | 3 | 4 | 5 | 6 | 7 |
| Breast | n (%) | 583 (59.1) | 73 (7.4) | 76 (7.7) | 53 (5.4) | 41 (4.2) | 28 (2.8) | 9 (0.9) | 124 (12.6) |
| n=987 | 95%CI | 56.0 to 62.1 | 5.9 to 9.2 | 6.2 to 9.5 | 4.1 to 6.9 | 3.0 to 5.5 | 1.9 to 4.0 | 0.5 to 1.7 | 10.6 to 14.7 |
| Prostate | n (%) | 512 (56.8) | 64 (7.1) | 101 (11.2) | 47 (5.2) | 37 (4.1) | 19 (2.1) | 12 (1.3) | 109 (12.1) |
| n=901 | 95%CI | 53.6 to 60.0 | 5.6 to 8.9 | 9.3 to 13.4 | 3.9 to 6.8 | 3.0 to 5.6 | 1.3 to 3.2 | 0.7 to 2.2 | 10.1 to 14.3 |
| Total | n (%) | 1095 (58.0) | 137 (7.3) | 177 (9.4) | 100 (5.3) | 78 (4.1) | 47 (2.5) | 21 (1.1) | 233 (12.3) |
| n=1888 | 95%CI | 55.8 to 60.2 | 6.2 to 8.5 | 8.1 to 10.8 | 4.4 to 6.4 | 3.3 to 4.1 | 1.9 to 3.3 | 0.7 to 1.7 | 10.9 to 13.9 |

'Which of the following term or terms had you heard of before?': n, %, 95 % CIs

| Screening information | Total | Not sure | Overdiagnosis | Overdetection | Unnecessary diagnosis | Overtreatment | Unnecessary treatment | False positive diagnosis | False positive test result |
|---|---|---|---|---|---|---|---|---|---|
| Breast | n (%) | 7 (0.7) | 215 (21.8) | 163 (16.5) | 214 (21.7) | 225 (22.8) | 280 (28.4) | 243 (24.6) | 270 (27.4) |
| n=987 | 95%CI | 0.3 to 1.4 | 19.3 to 24.4 | 14.3 to 18.9 | 19.2 to 24.3 | 20.3 to 25.5 | 25.6 to 31.2 | 22.0 to 27.4 | 24.6 to 30.2 |
| Prostate | n (%) | 14 (1.6) | 196 (21.8) | 137 (15.2) | 216 (24.0) | 186 (20.6) | 253 (28.1) | 232 (25.7) | 265 (29.4) |
| n=901 | 95%CI | 0.9 to 2.5 | 19.2 to 24.5 | 13.0 to 17.7 | 21.3 to 26.8 | 18.1 to 23.4 | 25.2 to 31.1 | 23.0 to 28.7 | 26.5 to 32.4 |
| Total | n (%) | 21 (1.1) | 411 (21.8) | 300 (15.9) | 430 (22.8) | 411 (21.8) | 533 (28.2) | 475 (25.2) | 535 (28.3) |
| n=1888 | 95%CI | 0.7 to 1.7 | 20.0 to 23.7 | 14.3 to 17.6 | 20.9 to 24.7 | 20.0 to 23.7 | 26.2 to 30.3 | 23.2 to 27.2 | 26.3 to 30.4 |

Participants receiving overdiagnosis information based on the Breast Screening Programme or the Prostate Cancer Risk Management Programme were analysed separately.

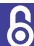

**Table 3** Familiarity with one or more versus no terms: descriptive statistics, adjusted ORs, 95% CIs and p values for variables in the multivariable binary logistic regression model

| Characteristic | Total (n=1888) | Familiarity with no terms versus at least one term: n (%) | | Adjusted OR, 95% CI | P values |
| --- | --- | --- | --- | --- | --- |
| | | No term (n=1095; 58.0 %) | At least one term (n=793; 42.0 %) | At least one familiar term (vs none) | |
| Overdiagnosis information | | | | | |
| Breast screening text | 987 | 583 (59.1) | 404 (40.9) | 0.87, 0.70 to 1.09 | 0.224 |
| versus prostate screening text | 901 | 512 (56.8) | 389 (43.2) | | |
| Gender | | | | | |
| Male | 879 | 534 (60.8) | 345 (39.2) | 1.00, 0.79 to 1.26 | 0.998 |
| versus female | 1009 | 561 (55.6) | 448 (44.4) | | |
| Ethnicity | | | | | |
| White British | 1432 | 775 (54.1) | 657 (45.9) | 2.14, 1.61 to 2.84 | **<0.0005** |
| versus other ethnic groups | 456 | 320 (70.2) | 136 (29.8) | | |
| Marital status | | | | | |
| Married or living as a couple | 1133 | 661 (58.3) | 472 (41.7) | 0.87, 0.69 to 1.09 | 0.216 |
| versus single, widowed, divorced or separated | 755 | 434 (57.5) | 321 (42.5) | | |
| Highest level of education | | | | | Overall: **<0.0005** |
| Other levels | 199 | 116 (58.3) | 83 (41.7) | 2.32, 1.46 to 3.68 | **<0.0005** |
| Approximately level 4 | 507 | 219 (43.2) | 288 (56.8) | 4.17, 2.75 to 6.33 | **<0.0005** |
| Approximately level 1, 2 or 3 | 890 | 532 (59.8) | 358 (40.2) | 2.19, 1.52 to 3.15 | **<0.0005** |
| versus no formal qualifications or don't know | 292 | 228 (78.1) | 64 (21.9) | | |
| Social class grade | | | | | Overall: **0.050** |
| Grade A or B | 386 | 170 (44.0) | 216 (56.0) | 1.54, 1.09 to 2.17 | **0.014** |
| Grade C1 or C2 | 931 | 542 (58.2) | 389 (41.8) | 1.19, 0.91 to 1.55 | 0.201 |
| versus grade D or E | 571 | 383 (67.1) | 188 (32.9) | | |
| Personal diagnosis of cancer | | | | | |
| Yes | 98 | 49 (50.0) | 49 (50.0) | 1.10, 0.68 to 1.80 | 0.690 |
| versus no | 1790 | 1046 (58.4) | 744 (41.6) | | |
| Knows someone with cancer | | | | | |
| Yes | 1113 | 562 (50.5) | 551 (49.5) | 1.50, 1.19 to 1.89 | **0.001** |
| versus no or don't know | 775 | 533 (68.8) | 242 (31.2) | | |
| Previously read a screening leaflet | | | | | |
| Yes | 1058 | 535 (50.6) | 523 (49.4) | 1.27, 0.97 to 1.66 | 0.089 |
| versus no or not sure | 830 | 560 (67.5) | 270 (32.5) | | |
| Previously read an NHS screening website | | | | | |
| Yes | 324 | 125 (38.6) | 199 (61.4) | 1.45, 1.07 to 1.97 | **0.016** |
| versus no or not sure | 1564 | 970 (62.0) | 594 (38.0) | | |
| Discussed screening with doctor/nurse | | | | | |
| Yes | 624 | 276 (44.2) | 348 (55.8) | 1.34, 1.03 to 1.74 | **0.030** |
| versus no | 1264 | 819 (64.8) | 445 (35.2) | | |
| Previously read or heard similar information | | | | | |
| Yes | 662 | 266 (40.2) | 396 (59.8) | 2.30, 1.81 to 2.94 | **<0.0005** |
| versus no or not sure | 1226 | 829 (67.6) | 397 (32.4) | | |
| Endorsed any term(s) as making sense | | | | | |

Continued

**Table 3** Continued

| Characteristic | Total (n=1888) | No term (n=1095; 58.0 %) | At least one term (n=793; 42.0 %) | At least one familiar term (vs none) | P values |
|---|---|---|---|---|---|
| | | **Familiarity with no terms versus at least one term: n (%)** | | **Adjusted OR, 95% CI** | |
| Yes | 1051 | 426 (40.5) | 625 (59.5) | 4.92, 3.94 to 6.17 | **<0.0005** |
| No | 837 | 669 (79.9) | 168 (20.1) | | |
| Perceived clarity of information | | | | Overall: 0.534 | |
| Extremely clear | 170 | 87 (51.2) | 83 (48.8) | 0.85, 0.52 to 1.40 | 0.525 |
| Very clear | 636 | 339 (53.3) | 297 (46.7) | 0.97, 0.65 to 1.43 | 0.860 |
| Moderately clear | 685 | 424 (61.9) | 261 (38.1) | 0.78, 0.53 to 1.16 | 0.225 |
| Slightly clear | 206 | 130 (63.1) | 76 (36.9) | 0.92, 0.57 to 1.49 | 0.745 |
| versus not at all clear | 191 | 115 (60.2) | 76 (39.8) | | |
| | | **Mean (SD)** | | | |
| Age (in years) | 1888 | 43.8 (15.7) | 43.3 (15.8) | 0.99, 0.98 to 1.00 | **0.003** |

Note: adjusted ORs and 95% CIs are relative to a stated reference category except age, which is per unit increase; p values <0.05 are in bold; all predictor variables are included in the model.
NHS, National Health Service.

education and age: white British participants were more likely to be familiar with at least one term compared with other ethnic groups. Participants with a higher (or 'other') level of education were also more likely to be familiar with one or more terms compared with participants who had no formal qualifications (or did not know). Older participants were less likely to be familiar with one or more terms. Participants were also more likely to be familiar with one or more terms if they knew someone with cancer, had previously read an NHS website about screening, discussed screening with a doctor or nurse, had previously read or seen similar information or endorsed one or more terms as making sense as labels.

Finally, there was moderate evidence against the null hypothesis for an association with social class grade with participants in grades A or B being more likely to be familiar with at least one term compared with participants in grades D or E.

### Terms endorsed as making sense

Table 4 shows that participants most commonly did not endorse any terms as making sense (44.3%). As with results for familiarity, the second most common number of terms endorsed was all seven (15.7%). The most commonly endorsed term was 'unnecessary treatment' (40.4% of participants in the sample endorsed this term overall; 41.4% and 39.3% after being given the breast and prostate information, respectively). Endorsement of other terms ranged from 27.9% ('overdetection') to 35.4% ('unnecessary diagnosis') for information from breast screening and 27.4% ('overdetection') to 36.6% ('false positive test results') for information from prostate screening.

Other than familiarity (table 3), the logistic regression analysis of predictors of endorsement only showed strong evidence against the null hypothesis for gender

and knowing someone with cancer: males and participants who knew someone diagnosed with cancer were more likely to endorse at least one term as making sense (table 5).

### DISCUSSION

This study aimed to address a gap in the literature regarding how best to communicate the issue of overdiagnosis.[8] We investigated whether people are familiar with terms used to define overdiagnosis or related concepts, and whether these made sense as labels for either of two widely used descriptions. We found that a majority of participants (58.0%) had never encountered any of the assessed terms before and a large proportion (44.3%) did not rate any as making sense as a suitable label. In addition, no specific term(s) emerged as being notably more familiar or applicable.

An earlier similar study found that only 30.0% of the UK general public (aged 50–70 years) had previously encountered the specific term, 'overdiagnosis',[9] which is broadly comparable with the 21.8% who reported having encountered it in this study. Our findings are also consistent with those of a previous study carried out as part of the redesign of the information materials for the NHS Breast Screening Programme.[20] Women were interviewed with the aim of finding ways to communicate the concept of overdiagnosis: participants generally struggled to understand it, regardless of whether it was labelled as 'overdiagnosis' or 'overtreatment'.

Terminology can have a role in transferring information to lay individuals as part of a description of technical concepts.[11 21] However, the main implication of our results is that there may currently only be

**Table 4** Number of participants (and percentages and 95% CIs) endorsing (1) each number of possible terms and (2) each specific term as making sense to them as a label

| Screening information | Total | Number of terms endorsed as making sense as a way of describing the text: n, %, 95% CIs | | | | | | | |
|---|---|---|---|---|---|---|---|---|---|
| | | 0 | 1 | 2 | 3 | 4 | 5 | 6 | 7 |
| Breast | n (%) | 429 (43.5) | 76 (7.7) | 86 (8.7) | 74 (7.5) | 67 (6.8) | 58 (5.9) | 42 (4.3) | 155 (15.7) |
| n=987 | 95% CI | 40.4 to 46.6 | 6.2 to 9.5 | 7.1 to 10.6 | 6.0 to 9.3 | 5.3 to 8.5 | 4.5 to 7.5 | 3.1 to 5.7 | 13.5 to 18.1 |
| Prostate | n (%) | 408 (45.3) | 59 (6.5) | 73 (8.1) | 68 (7.5) | 66 (7.3) | 48 (5.3) | 37 (4.1) | 142 (15.8) |
| n=901 | 95% CI | 42.1 to 48.5 | 5.1 to 8.3 | 6.5 to 10.0 | 6.0 to 9.4 | 5.8 to 9.2 | 4.0 to 6.9 | 3.0 to 5.6 | 13.5 to 18.2 |
| Total | n (%) | 837 (44.3) | 135 (7.2) | 159 (8.4) | 142 (7.5) | 133 (7.0) | 106 (5.6) | 79 (4.2) | 297 (15.7) |
| n=1888 | 95% CI | 42.1 to 46.6 | 6.1 to 8.4 | 7.2 to 9.7 | 6.4 to 8.8 | 6.0 to 8.3 | 4.6 to 6.7 | 3.4 to 5.2 | 14.1 to 17.4 |

| Screening information | Total | 'Which terms make sense as a way of describing this risk of a screening test?': n, %, 95% CIs | | | | | | |
|---|---|---|---|---|---|---|---|---|
| | | Overdiagnosis | Overdetection | Unnecessary diagnosis | Overtreatment | Unnecessary treatment | False positive diagnosis | False positive test results |
| Breast | n (%) | 334 (33.8) | 275 (27.9) | 349 (35.4) | 336 (34.0) | 409 (41.4) | 324 (32.8) | 338 (34.2) |
| n=987 | 95% CI | 30.9 to 36.8 | 25.1 to 30.7 | 32.4 to 38.4 | 31.1 to 37.0 | 38.4 to 44.5 | 30.0 to 35.8 | 31.3 to 37.2 |
| Prostate | n (%) | 291 (32.3) | 247 (27.4) | 307 (34.1) | 280 (31.1) | 354 (39.3) | 320 (35.5) | 330 (36.6) |
| n=901 | 95% CI | 29.3 to 35.4 | 24.6 to 30.4 | 31.0 to 37.2 | 28.1 to 34.2 | 36.1 to 42.5 | 32.4 to 38.7 | 33.5 to 39.8 |
| Total | n (%) | 625 (33.1) | 522 (27.6) | 656 (34.7) | 616 (32.6) | 763 (40.4) | 644 (34.1) | 668 (35.4) |
| n=1888 | 95% CI | 31.0 to 35.3 | 25.7 to 29.7 | 32.6 to 36.9 | 30.5 to 34.8 | 38.2 to 42.6 | 32.0 to 36.3 | 33.2 to 37.6 |

**Table 5** Endorsed one or more versus no terms as making sense: descriptive statistics, adjusted ORs, 95% CIs and p values for variables in the multivariable binary logistic regression model

| Characteristic | Total (n=1888) | Endorsed no versus at least one term as making sense: n (%) | | Adjusted OR, 95% CI | P value |
| | | No term (n=837; 44.3 %) | At least one term (n=1051; 55.7%) | At least one term endorsed (versus no terms endorsed) | |
| --- | --- | --- | --- | --- | --- |
| Overdiagnosis information | | | | | |
| Breast screening text | 987 | 429 (43.5) | 558 (56.5) | 1.16, 0.95 to 1.42 | 0.148 |
| versus prostate screening text | 901 | 408 (45.3) | 493 (54.7) | | |
| Gender | | | | | |
| Male | 879 | 380 (43.2) | 499 (56.8) | 1.40, 1.13 to 1.74 | **0.002** |
| versus female | 1009 | 457 (45.3) | 552 (54.7) | | |
| Ethnicity | | | | | |
| white British | 1432 | 592 (41.3) | 840 (58.7) | 1.25, 0.97 to 1.61 | 0.080 |
| versus other ethnic groups | 456 | 245 (53.7) | 211 (46.3) | | |
| Marital status | | | | | |
| Married or living as a couple | 1133 | 500 (44.1) | 633 (55.9) | 1.05, 0.85 to 1.30 | 0.671 |
| versus single, widowed, divorced or separated | 755 | 337 (44.6) | 418 (55.4) | | |
| Highest level of education | | | | Overall: 0.556 | |
| Other levels | 199 | 88 (44.2) | 111 (55.8) | 1.21, 0.80 to 1.82 | 0.374 |
| Approximately level 4 | 507 | 190 (37.5) | 317 (62.5) | 1.27, 0.88 to 1.84 | 0.204 |
| Approximately level 1, 2 or 3 | 890 | 396 (44.5) | 494 (55.5) | 1.24, 0.91 to 1.70 | 0.169 |
| versus no formal qualifications or don't know | 292 | 163 (55.8) | 129 (44.2) | | |
| Social class grade | | | | Overall: 0.574 | |
| Grade A or B | 386 | 137 (35.5) | 249 (64.5) | 1.09, 0.78 to 1.51 | 0.617 |
| Grade C1 or C2 | 931 | 420 (45.1) | 511 (54.9) | 0.94, 0.74 to 1.19 | 0.604 |
| versus grade D or E | 571 | 280 (49.0) | 291 (51.0) | | |
| Personal diagnosis of cancer | | | | | |
| Yes | 98 | 37 (37.8) | 61 (62.2) | 1.08, 0.68 to 1.74 | 0.741 |
| versus no | 1790 | 800 (44.7) | 990 (55.3) | | |
| Knows someone with cancer | | | | | |
| Yes | 1113 | 427 (38.4) | 686 (61.6) | 1.35, 1.09 to 1.68 | **0.005** |
| versus no or don't know | 775 | 410 (52.9) | 365 (47.1) | | |
| Previously read a screening leaflet | | | | | |
| Yes | 1058 | 416 (39.3) | 642 (60.7) | 1.18, 0.91 to 1.52 | 0.206 |
| versus no or not sure | 830 | 421 (50.7) | 409 (49.3) | | |
| Previously read an NHS screening website | | | | | |
| Yes | 324 | 109 (33.6) | 215 (66.4) | 1.01, 0.75 to 1.37 | 0.955 |
| versus no or not sure | 1564 | 728 (46.5) | 836 (53.5) | | |
| Discussed screening with doctor/nurse | | | | | |
| Yes | 624 | 217 (34.8) | 407 (65.2) | 1.25, 0.97 to 1.61 | 0.091 |
| versus no | 1264 | 620 (49.1) | 644 (50.9) | | |
| Previously read or heard similar information | | | | | |
| Yes | 662 | 224 (33.8) | 438 (66.2) | 1.18, 0.93 to 1.50 | 0.185 |
| versus no or not sure | 1226 | 613 (50.0) | 613 (50.0) | | |
| Previously encountered any term(s) | | | | | |

Continued

**Table 5** Continued

| Characteristic | Total (n=1888) | Endorsed no versus at least one term as making sense: n (%) | | Adjusted OR, 95% CI | P value |
|---|---|---|---|---|---|
| | | No term (n=837; 44.3 %) | At least one term (n=1051; 55.7%) | At least one term endorsed (versus no terms endorsed) | |
| Yes | 1051 | 168 (21.2) | 625 (78.8) | 4.88, 3.89 to 6.11 | **<0.0005** |
| No | 837 | 669 (61.1) | 426 (38.9) | | |
| Perceived clarity of information | | | | Overall: 0.148 | |
| Extremely clear | 170 | 67 (39.4) | 103 (60.6) | 1.36, 0.85 to 2.17 | 0.204 |
| Very clear | 636 | 255 (40.1) | 381 (59.9) | 1.38, 0.96 to 1.98 | 0.083 |
| Moderately clear | 685 | 308 (45.0) | 377 (55.0) | 1.39, 0.97 to 1.98 | 0.073 |
| Slightly clear | 206 | 109 (52.9) | 97 (47.1) | 0.99, 0.64 to 1.53 | 0.953 |
| versus not at all clear | 191 | 98 (51.3) | 93 (48.7) | | |
| | | **Mean (SD)** | | | |
| Age (in years) | 1888 | 43.4 (15.6) | 43.7 (15.8) | 1.00, 0.99 to 1.00 | 0.262 |

Note: adjusted ORs and 95% CIs are relative to a stated reference category except age, which is per unit increase; p values <0.05 are in bold; all predictor variables are included in the model.

limited value in using any of the labels that we tested. Terms used were neither familiar nor rated as particularly relevant in this large, diverse sample, meaning that they may be confusing, unintuitive and hard for people to memorise and recall. Even the most technically appropriate terms (eg, 'overdiagnosis' and 'overdetection') were not endorsed as making sense more frequently than suggested alternative terminology (eg, 'unnecessary diagnosis') or more inaccurate and potentially problematic terminology (eg, 'false positive diagnosis'). To some extent, the currently limited familiarity and low levels of endorsement supports UK policy decisions not to use a specific term in the revised breast screening information leaflet.[20] Furthermore, shortly before the start of recruitment, an updated NHS information leaflet regarding PSA screening was published, with additional detail on overdiagnosis and overtreatment[22]; this also omitted any specific descriptive label.

A potential explanation for the low proportions of participants endorsing each term comes from a recent survey in the USA. This used a similar design to the present study in which women aged 35–55 years were recruited to respond to brief descriptions of overdiagnosis and overtreatment. Only a small minority of participants (approximately 15%–20%) endorsed messages as either believable and something with which they agreed.[23] To some extent, this is consistent with qualitative research showing that the concept is often surprising and confusing to lay people.[6 7] Limited acceptance of the description of overdiagnosis itself may have resulted in participants disengaging with the concept and hence not rating any of the available terms as appropriate. It may be relevant that both sets of descriptions were presented outside of contexts in which they would normally be encountered (eg, the original screening information

leaflets from which they were adapted). Both comprehension and acceptance may be different (either better or worse) in the presence of additional detail,[15] either from a full leaflet or as part of a discussion with a clinician. This warrants further research.

One unexpected finding was that the second most common number of terms that were familiar or endorsed was all seven (12.3% and 15.7%, respectively). This may be due to the differences between terms being subtle: members of the public may not recall exactly which terms they had encountered but felt that broader concepts like diagnosis and treatment were familiar.

We found that participants with a lower level of education, who were not white British, were older or (possibly) were from a lower social class grade were less likely to have previously encountered any term(s), suggesting that these groups may be particularly poorly served by the use of any specific terminology. Unsurprisingly, those who had previous exposure to cancer (ie, by knowing someone with a diagnosis) and those who had encountered potentially relevant information (eg, those who had previously discussed screening with a doctor or nurse) were more likely to have encountered any term(s). In some respects, this latter finding was surprising because although medical staff might be expected to discuss overdiagnosis with patients, there is evidence that the topic is explained only rarely.[24] Those who knew someone with cancer were also more likely to endorse at least one term as making sense. It is unclear why males were more likely to endorse term(s) as making sense. This may warrant further research (possible explanations may include a degree of overconfidence in men or underconfidence in women), although this should be considered in the context of small absolute differences observed here (56.8% vs 54.7% endorsing at least one term).

Continuing communication efforts mean that these results may change over time; it is plausible that ongoing campaigns (eg, refs [3–5]) will result in the public becoming increasingly familiar with the concept and terminology, having a better understanding of the differences between similar terms (eg, seeing *'false positives'* as distinct from *'overdiagnosis'*) and being more likely to endorse particular terms as applicable. However, present attempts to communicate overdiagnosis (eg, as part of screening invitations or mainstream media stories) may be more effective if explicit descriptions are used, such as adapted forms of information from the Breast Screening Programme. This has undergone an extensive design process[20] and is more likely to be rated as clearer than the equivalent from the Prostate Cancer Risk Management Programme. However, fewer than half of participants (46.6%) rated it as very or extremely clear, suggesting that future research could be used to improve it further (eg, by providing additional detail).[15]

This study has limitations. The data collection method used by the survey company meant that demographic characteristics used to ensure population representativeness were not exhaustive (eg, they omitted ethnicity and education), and response rates were not available. The list of terms tested was also not exhaustive; there may be suitable terminology that we did not assess. However, we found very similar levels of endorsement for terms that used several variations on the themes of diagnosis, treatment and false positives, providing some evidence that any superior terms would have to be quite different.

### Conclusions

This study tested whether alternative terminology to 'overdiagnosis' has the potential to help increase awareness and understanding of the concept among individuals who may face healthcare decisions that would put them at risk. These results suggest that, at present, using specific terms would have limited benefits and may be less well suited to particular groups (eg, less educated or non-white British individuals). It may currently be more effective to refer to the concept via more explicit descriptions. Previous research indicates that information from the NHS Breast Screening Programme may be a relatively effective template for this purpose. However, future research could explore scope for improvement.

**Contributors** All authors conceived and designed the study. AG analysed the data. All authors participated in the interpretation of results. All authors drafted the manuscript, participated in critical revision and approved the final version.

**Funding** This work was supported by a programme grant from Cancer Research UK awarded to Professor Jane Wardle (C1418/A14134). Jo Waller is supported by a Career Development Fellowship from Cancer Research UK (C7492/A17219).

**Disclaimer** Cancer Research UK was not involved in the design of this study; the collection, analysis, or interpretation of the results; in the writing of the manuscript; or in the decision to submit for publication.

**Competing interests** None declared.

**Patient consent** Not required.

**Ethics approval** Institutional approval was granted by the University College London Research Ethics Committee (5771/002).

**Provenance and peer review** Not commissioned; externally peer reviewed.

**Data sharing statement** No additional data are available.

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
