## [Reviewer comments · BMJ Open]

ARTICLE DETAILS

TITLE (PROVISIONAL)	Improving public understanding of ‘overdiagnosis’ in England: a population survey assessing familiarity with possible terms for labelling the concept and perceptions of appropriate terminology
AUTHORS	Ghanouni, Alex; Renzi, Cristina; Waller, Jo

VERSION 1 – REVIEW

REVIEWER	Karen Born Institute of Health Policy, Management & Evaluation, Faculty of Medicine, University of Toronto, Canada
REVIEW RETURNED	09-Jan-2018

GENERAL COMMENTS	Thanks for the opportunity to review this manuscript. It addresses an important gap in the literature and certainly provides opportunities for additional scholarship to build upon findings. In particular a more in depth qualitative analysis would be merited to better understand public perspectives. Specific comments are below: Introduction - The first sentence is very unclear and should be removed or reworded. Who holds this growing concern? The public? Participants - were the 3 demographic characteristics the only ones collected? What about ethnicity and education? These are certainly not exhaustive for representativeness and could be seen as a limitation, or acknowledged as one. For Table 2 Were they asked to recall if they had ever heard terms before if or if they had heard within a certain period of time?
---

REVIEWER	Dr. Prim. a.D. Hans Concini (1), Prof. Dr. Gabriele Nagel, MPH (2), Kristin Ganahl, MA (1) (1) Agency for Preventive and Social Medicine Vorarlberg, Austria (2) University of Ulm, Department for Epidemiology and Medical Biometry
REVIEW RETURNED	18-Jan-2018

GENERAL COMMENTS	Reviewer Comment to „Improving public understanding of ‘overdiagnosis’: a population survey assessing familiarity with possible term for labeling the concept and perception of appropriate terminology” We sincerely congratulate the authors on a paper that is remarkable both in terms of novelty and relevance. As far as we know, this is the
--

first paper of its kind. The aim of this article is to address how to best communicate the concept of overdiagnosis by displaying results from the first observational study in UK on this topic. This topic is of great interest considering the increasing debate on the significance and implications of overdiagnosis. The authors describe the study research question, and used research design as well as sampling procedure clearly and comprehensively. Considering the described sampling procedure and the representative national sample, this study has great potential to answer the important research question on how to better communicate the term 'overdiagnosis'. The manuscript is well written and easy to follow.

Nevertheless we would like to bring a few points to the attention of the authors:

We think this paper as a whole would benefit from a more refined analysis in order to learn more about the different groups in society and how the term overdiagnosis is understood or rather misunderstood. For example group comparisons – considering the demographic variables (i.e. gender, age, education, social class, etc.) or multivariate analysis could contribute to a better understanding of which groups in society understand the term 'overdiagnosis' and who misunderstands the term. More specific results could offer more precise implication on how the term could be better communicated to the general population, and if the different demographic groups in society should be addressed differently. It would also be interesting to see how the variables on "familiarity with screening information" relate to the understanding of the term 'overdiagnosis'. We think that more detailed analysis could lead to more precise results to better address the initial research question on how the concept of overdiagnosis could be communicated better.

Furthermore, the authors used the information leaflets of the Breast Cancer Screening Programme and Prostate Cancer Management Programme. Since not all readers are familiar with these programmes, it would be helpful, to add some basic information about the programmes.

In summary, we congratulate the authors for bringing attention to this important and unique research topic.

Prim. a.D. Dr. Hans Concin

Agency for Preventive and Social Medicine

Prof. Dr. Gabriele Nagel, MPH
University of Ulm, Department for Epidemiology and Medical Biometry

Kristin Ganahl, MA

	Agency for Preventive and Social Medicine
--	---

REVIEWER	Rebekah Nagler Assistant Professor Hubbard School of Journalism & Mass Communication University of Minnesota United States
REVIEW RETURNED	27-Jan-2018

GENERAL COMMENTS	This study examines public familiarity with and endorsement of terms used to describe overdiagnosis. Although this is a well-written discussion of an important topic, I do have some questions about the study's contribution to the growing literature. These questions, along with some additional points, are summarized below. 1. The authors suggest that this paper is helping to fill a gap in the literature—namely, we do not know how best to communicate about overdiagnosis. I agree that this is a real concern, and yet I'm not sure if this paper really contributes to our understanding here. The authors suggest that the term "overdiagnosis" may be confusing, and so they sought to assess whether alternative descriptions might be preferable. But my understanding of the literature in this area—including previous studies published by these authors (see, for example, citations 7 and 9) and others (citations 6 and 11)—is that we already know that actual understanding of this term is limited. The aforementioned prior qualitative and textual analyses have suggested as much, which is why a recent study attempted to assess awareness of and reactions to a statement about (i.e., definition of) overdiagnosis (see Nagler RH, Fowler EF, Gollust SE, Women's awareness of and responses to messages about breast cancer overdiagnosis and overtreatment: Results from a 2016 national survey. Medical Care. 2017; 55(10): 879-885). Although I suppose it is conceivable that terms other than "overdiagnosis" might be more readily understood, I'm not sure there is strong theoretical reason to believe this would be the case. In short, the argument and motivation for this study—and how it moves the field forward—needs greater development. 2. The authors removed references to cancer from the NHS leaflet prompts so that findings could be generalized beyond the breast and prostate cancer screening contexts presented in those leaflets. Although I can understand the line of thinking here, I do wonder if it might have made it even more difficult for participants to make sense of this information/the terms because there was no anchoring information—just a vague reference to "screening tests to check for illnesses." 3. Table 1 offers substantial demographic and clinical information on participants, some of which we know to be important predictors of awareness of and receptivity to overdiagnosis and overtreatment arguments (see Nagler et al. study). Did the authors consider trying to explain the observed variation in familiarity with and endorsement of various terms (i.e., antecedents of familiarity and endorsement)? A few methodological questions: 4. I know descriptions of the survey and measures were published elsewhere, but I think the manuscript would benefit from a little more methodological detail (so that readers don't have to look
--

	up another study in order to understand the current study's measures, etc.). As far as the sample is concerned, it would be useful to have response rate and/or cooperation rate information. 5. I do wonder about the endorsement measure's construct validity. Has this measure been used previously? If not, can the authors say something about how they developed the measure, and whether there is any evidence that it captures what it's intended to capture? One could imagine participants interpreting it in distinct ways.... 6. The authors indicate that this study uses items from a larger survey (the ABACUS survey). To the extent that this survey included additional information on overdiagnosis and related concepts, it would be helpful to hear more about this content and the order in which questions appeared. It seems plausible that responses to earlier questions could have primed (or otherwise shaped) responses to the items assessed here.
--	---

VERSION 1 – AUTHOR RESPONSE

REVIEWERS' COMMENTS TO THE AUTHORS:

REVIEWER 1

Reviewer Name: Karen Born

Institution and Country: Institute of Health Policy, Management & Evaluation, Faculty of Medicine, University of Toronto, Canada

Competing Interests: I am the knowledge translation lead for Choosing Wisely Canada.

Thanks for the opportunity to review this manuscript. It addresses an important gap in the literature and certainly provides opportunities for additional scholarship to build upon findings. In particular a more in depth qualitative analysis would be merited to better understand public perspectives.

AR: We agree with this point and reiterate that part of the impetus for this study came from findings in previous qualitative studies (e.g. 6,7). We agree that further qualitative work would be valuable, particularly as a means of generating potential communication strategies that could be tested in future quantitative studies.

Introduction - The first sentence is very unclear and should be removed or reworded. Who holds this growing concern? The public?

AR: We have amended the first sentence to indicate this concern exists primarily among healthcare providers and policymakers.

Participants - Were the 3 demographic characteristics the only ones collected? What about ethnicity and education? These are certainly not exhaustive for representativeness and could be seen as a limitation, or acknowledged as one.

AR: We agree that the attributes used with the aim of achieving representativeness are not exhaustive. We have amended this section to provide the complete list of attributes and acknowledged this as a limitation in the Discussion.

Results - For Table 2, were they asked to recall if they had ever heard terms before if or if they had heard within a certain period of time?

AR: We refer to the Measures: Main outcomes section in which we quote the wording of the relevant question: “The second outcome measure (familiarity) was: “Have you ever seen or heard any of the previous seven terms before today?””.

REVIEWER 2

Reviewer Name: Dr. Prim. A.D. Hans Concin (1), Prof. Dr. Gabriele Nagel, MPH (2), Kristin Ganahl, MA (1)

Institution and Country: (1) Agency for Preventive and Social Medicine Vorarlberg, Austria; (2)

University of Ulm, Department for Epidemiology and Medical Biometry

Competing Interests: None declared

We sincerely congratulate the authors on a paper that is remarkable both in terms of novelty and relevance. As far as we know, this is the first paper of its kind. The aim of this article is to address how to best communicate the concept of overdiagnosis by displaying results from the first observational study in UK on this topic. This topic is of great interest considering the increasing debate on the significance and implications of overdiagnosis. The authors describe the study research question, and used research design as well as sampling procedure clearly and comprehensively. Considering the described sampling procedure and the representative national sample, this study has great potential to answer the important research question on how to better communicate the term ‘overdiagnosis’. The manuscript is well written and easy to follow.

AR: We thank the reviewers for their positive comments regarding the originality and significance of the study, and quality of the manuscript.

Nevertheless, we would like to bring a few points to the attention of the authors:

We think this paper as a whole would benefit from a more refined analysis in order to learn more about the different groups in society and how the term overdiagnosis is understood or rather misunderstood. For example group comparisons – considering the demographic variables (i.e. gender, age, education, social class, etc.) or multivariate analysis could contribute to a better understanding of which groups in society understand the term ‘overdiagnosis’ and who misunderstands the term. More specific results could offer more precise implication on how the term could be better communicated to the general population, and if the different demographic groups in society should be addressed differently.

AR: We agree that analyses of familiarity with and endorsement of terms are relevant. Indeed, prior to submission, previous iterations of the manuscript included additional exploratory analyses broadly consistent with the reviewers’ recommendation. These were omitted for the initial submission for the purposes of succinctness and in order to focus on the main study findings (i.e. that familiarity with/endorsement of terms was generally low without a clearly superior label). Given the strong endorsement of multivariable analyses by Reviewers 2 and Reviewer 3, we have reinstated a revised version of them and amended the paper accordingly.

The analyses assess predictors of i) familiarity with at least one term and ii) endorsement of at least one term as making sense, rather than testing familiarity with/endorsement of specific terms such as ‘overdiagnosis’. We made this decision for two reasons. First, the distribution of results was essentially bimodal for both familiarity and endorsement (e.g. most participants either endorsed none of the suggested terms, or endorsed all of the suggested terms). Similarly, responses for each term were highly correlated. For example, among participants who endorsed ‘overdiagnosis’ as making sense as a label for the given description, the proportions who also endorsed each of the other specific terms as making sense (i.e. from ‘overdetection’ to ‘false positive test results’) ranged from 69% to 82%. Conversely, among participants who did not endorse ‘overdiagnosis’ as making sense

as a label (i.e. responding “no” or “not sure”), the large majority did not endorse any other given term; concordance ranged from 80.4% to 93.0%.

In our view, this suggests there was relatively little justification for assessing predictors of familiarity with/endorsement of any specific term(s). Instead, we think it is more meaningful to assess predictors of familiarity/endorsement in aggregate.

It would also be interesting to see how the variables on “familiarity with screening information” relate to the understanding of the term ‘overdiagnosis’. We think that more detailed analysis could lead to more precise results to better address the initial research question on how the concept of overdiagnosis could be communicated better.

AR: We agree with this suggestion; we have included predictor variables on familiarity with screening information as part of the analyses described above (e.g. having previously read a screening leaflet, or read or heard similar information).

Furthermore, the authors used the information leaflets of the Breast Cancer Screening Programme and Prostate Cancer Management Programme. Since not all readers are familiar with these programmes, it would be helpful, to add some basic information about the programmes.

AR: We have amended the Methods: Design section to provide additional detail on the nature of these Programmes.

In summary, we congratulate the authors for bringing attention to this important and unique research topic.

Prim. A.D. Dr. Hans Concin
Agency for Preventive and Social Medicine

Prof. Dr. Gabriele Nagel, MPH
University of Ulm, Department for Epidemiology and Medical Biometry

Kristin Ganahl, MA
Agency for Preventive and Social Medicine

REVIEWER 3

Reviewer Name: Rebekah Nagler
Institution and Country: Assistant Professor, Hubbard School of Journalism & Mass Communication, University of Minnesota, United States
Competing Interests: None declared

This study examines public familiarity with and endorsement of terms used to describe overdiagnosis. Although this is a well-written discussion of an important topic, I do have some questions about the study’s contribution to the growing literature. These questions, along with some additional points, are summarized below.

AR: We thank the reviewer for their comments on the quality of writing and topic. We have responded to their critiques below.

1. The authors suggest that this paper is helping to fill a gap in the literature—namely, we do not know how best to communicate about overdiagnosis. I agree that this is a real concern, and yet I'm not sure if this paper really contributes to our understanding here. The authors suggest that the term "overdiagnosis" may be confusing, and so they sought to assess whether alternative descriptions might be preferable. But my understanding of the literature in this area—including previous studies published by these authors (see, for example, citations 7 and 9) and others (citations 6 and 11)—is that we already know that actual understanding of this term is limited. The aforementioned prior qualitative and textual analyses have suggested as much, which is why a recent study attempted to assess awareness of and reactions to a statement about (i.e., definition of) overdiagnosis (see Nagler RH, Fowler EF, Gollust SE, Women's awareness of and responses to messages about breast cancer overdiagnosis and overtreatment: Results from a 2016 national survey. *Medical Care*. 2017; 55(10): 879-885). Although I suppose it is conceivable that terms other than "overdiagnosis" might be more readily understood, I'm not sure there is strong theoretical reason to believe this would be the case. In short, the argument and motivation for this study—and how it moves the field forward—needs greater development.

AR: We agree with the reviewer's interpretation of the literature with respect to findings that public understanding of the term is limited and accept that the justification for the present study was not elaborated on sufficiently. We have revised the introduction to describe some of the key findings from previous research, how these support the study premise, and how this study builds on previous work.

2. The authors removed references to cancer from the NHS leaflet prompts so that findings could be generalized beyond the breast and prostate cancer screening contexts presented in those leaflets. Although I can understand the line of thinking here, I do wonder if it might have made it even more difficult for participants to make sense of this information/the terms because there was no anchoring information—just a vague reference to "screening tests to check for illnesses."

AR: We agree that we are unable to rule out this possibility. However, we would not necessarily hypothesize that this is the case: previous studies have been consistent in demonstrating that information on overdiagnosis is challenging for the public to understand, irrespective of whether it is presented in the context of cancer screening (e.g. 11) or healthcare in general (e.g. 9, 10). This does not suggest that anchoring information around a specific disease would have made an appreciable difference to results.

3. Table 1 offers substantial demographic and clinical information on participants, some of which we know to be important predictors of awareness of and receptivity to overdiagnosis and overtreatment arguments (see Nagler et al. study). Did the authors consider trying to explain the observed variation in familiarity with and endorsement of various terms (i.e., antecedents of familiarity and endorsement)?

AR: Per our response to Reviewers 2, we have added exploratory analyses testing possible predictors of familiarity and endorsement.

A few methodological questions:

4. I know descriptions of the survey and measures were published elsewhere, but I think the manuscript would benefit from a little more methodological detail (so that readers don't have to look up another study in order to understand the current study's measures, etc.). As far as the sample is concerned, it would be useful to have response rate and/or cooperation rate information.

AR: We have added additional methodological detail in the methods. In particular, these include the eligibility for questions on screening participation and methods of coding predictor variables. We have

also added an additional limitation regarding the response rate: due to the recruitment method, this is not available.

5. I do wonder about the endorsement measure's construct validity. Has this measure been used previously? If not, can the authors say something about how they developed the measure, and whether there is any evidence that it captures what it's intended to capture? One could imagine participants interpreting it in distinct ways....

AR: This measure was developed for the present study. However, related to the previous comment, we have added additional detail regarding the development of the survey (specifically, the piloting with members of the population of interest, consisting of cognitive interviews followed by administering the full survey to a small sample online).

6. The authors indicate that this study uses items from a larger survey (the ABACUS survey). To the extent that this survey included additional information on overdiagnosis and related concepts, it would be helpful to hear more about this content and the order in which questions appeared. It seems plausible that responses to earlier questions could have primed (or otherwise shaped) responses to the items assessed here.

AR: We have added an additional section to the method (Ancillary measures) in which we describe the layout and measures used in the full ABACUS survey.

VERSION 2 – REVIEW

REVIEWER	Hans Concin (1), Gabriele Nagel (2), Kristin Ganahl (1) (1)Agency for Preventive and Social Medicine (2) Universtiy of Ulm. Department of Epidemiology and Medical Biometry
REVIEW RETURNED	22-Mar-2018

GENERAL COMMENTS	We congratulate the authors to the revisions and the improved quality of the article. Please consider adapting the box with strength and limitation of this study according to revisions
--

REVIEWER	Rebekah Nagler Hubbard School of Journalism & Mass Communication, University of Minnesota, United States
REVIEW RETURNED	08-Apr-2018

GENERAL COMMENTS	The authors have successfully responded to most comments and questions included in the initial review. Given the study findings, and in particular their central conclusion that "explicit descriptions [of overdiagnosis] may be more effective," I would be interested to hear the authors engage with our team's recent work on public reactions to such explicit descriptions (see citation included in initial review, and pasted below). In a population-based sample of U.S. women aged 35-55, we found that most women, once made aware of overdiagnosis and overtreatment, did not find such descriptions/statements to be believable or persuasive. Might such findings have implications for the communication about overdiagnosis? Could it be that explicit descriptions, in isolation (as we tested them in our study), are insufficient, and greater context is necessary? Or must such descriptions occur alongside patient-provider discussion? It seems that increasing awareness and recognition of these terms is not the only goal; it is also important for
---

	the public to understand these terms so that they can weigh potential harms of screening in their decision making. In the discussion, the authors might consider engaging with this -- not just how we can increase awareness and recognition, but ultimately promote greater understanding and receptivity. Nagler RH, Fowler EF, Gollust SE, Women's awareness of and responses to messages about breast cancer overdiagnosis and overtreatment: Results from a 2016 national survey. Medical Care. 2017; 55(10): 879-885
--	--

VERSION 2 – AUTHOR RESPONSE

REVIEWERS' COMMENTS TO THE AUTHORS:

REVIEWER 2

Reviewer Name: Hans Concin (1), Gabriele Nagel (2), Kristin Ganahl (1)
 Institution and Country: (1) Agency for Preventive and Social Medicine; (2) Universtiy of Ulm.
 Department of Epidemiology and Medical Biometry, Germany
 Competing Interests: None declared

We congratulate the authors to the revisions and the improved quality of the article. Please consider adapting the box with strength and limitation of this study according to revisions

Authors' Response: We thank the reviewers for their positive feedback on the amendments to the manuscript and for noting the scope to revise the box of strengths and limitations, which we have done.

REVIEWER 3

Reviewer Name: Rebekah Nagler
 Institution and Country: Hubbard School of Journalism & Mass Communication, University of Minnesota, United States
 Competing Interests: None declared

The authors have successfully responded to most comments and questions included in the initial review. Given the study findings, and in particular their central conclusion that "explicit descriptions [of overdiagnosis] may be more effective," I would be interested to hear the authors engage with our team's recent work on public reactions to such explicit descriptions (see citation included in initial review, and pasted below). In a population-based sample of U.S. women aged 35-55, we found that most women, once made aware of overdiagnosis and overtreatment, did not find such descriptions/statements to be believable or persuasive. Might such findings have implications for the communication about overdiagnosis? Could it be that explicit descriptions, in isolation (as we tested them in our study), are insufficient, and greater context is necessary? Or must such descriptions occur alongside patient-provider discussion? It seems that increasing awareness and recognition of these terms is not the only goal; it is also important for the public to understand these terms so that they can weigh potential harms of screening in their decision making. In the discussion, the authors might consider engaging with this -- not just how we can increase awareness and recognition, but ultimately promote greater understanding and receptivity.

Nagler RH, Fowler EF, Gollust SE, Women's awareness of and responses to messages about breast cancer overdiagnosis and overtreatment: Results from a 2016 national survey. *Medical Care*. 2017; 55(10): 879-885

AR: We thank the reviewer for highlighting their work, which is clearly highly relevant to our study. We have added a paragraph in the Discussion to cover this, in which we suggest that the generally low levels of endorsement of different terms in the present study may have been the result of limited belief in and agreement with information about the concept described. We have also moved a point that was previously described as a limitation to this paragraph, in which we acknowledge that comprehension (and, per Nagler et al., acceptance) of information on overdiagnosis/overtreatment may differ when provided with more context.

OTHER COMMENTS

Authors' note for the editorial team: We would like to point out that we have made minor edits for grammar and style throughout the manuscript. All changes have been tracked.